# Diversity and Structure of Soil Microbial Communities in Chinese Fir Plantations and *Cunninghamia lanceolata–Phoebe bournei* Mixed Forests at Different Successional Stages

Weiyang Li [1,†], Huimin Sun [2,†], Minmin Cao [1], Liyan Wang [3], Xianghua Fang [4] and Jiang Jiang [1,*]

1 Co-Innovation Center for Sustainable Forestry in Southern China, Jiangsu Province Key Laboratory of Soil and Water Conservation and Ecological Restoration, Nanjing Forestry University, 159 Longpan Road, Nanjing 210037, China; liweiyang2023@outlook.com (W.L.); healer_cao@outlook.com (M.C.)
2 Center for Eco-Environmental Research, Nanjing Hydraulic Research Institute, Nanjing 210029, China; huimin_sun@outlook.com
3 Jiangxi Academy of Forestry and Nanchang Urban Ecosystem Research Station, Nanchang 330013, China; wangliyan052@163.com
4 Longquan Nature Conservation Centre, Hundred Mountains Group National Forest Park, Longquan 323700, China; selina_cui@outlook.com
* Correspondence: ecologyjiang@gmail.com
† These authors contributed equally to this work.

**Abstract:** *Cunninghamia lanceolata* is an important species in plantations and is widely planted in sub-tropical regions of China because of its fast-growing and productive characteristics. However, the monoculture planting is carried out in the pursuit of economic value. This planting mode has led to problems such as the exhaustion of soil fertility, decrease in vegetation diversity, and decrease in woodland productivity. In order to restore soil fertility and increase timber production, the introduction of broad-leaved tree species to plantations is an effective transformation model. Understanding how forest age changes and stand structure differences drive the composition and diversity of soil microbial communities is helpful in understanding the trend of soil–microbial changes in plantations and evaluating the effects of the introduction of broad-leaved tree species in soil–plant–microbial ecosystems in plantations. Therefore, the purpose of our study is to investigate the effects of forest age and pure forest conversion on *C. lanceolata–P. bournei*-mixed forest soil microbial community structure and diversity by detecting soil nutrients, enzyme activities, and soil microbial 16S and ITS rRNA gene sequencing. According to the findings, the diversity and abundance of bacterial communities in *C. lanceolata* plantations of different ages increased first and then decreased with the increase in forest age, and the max value was in the near-mature forest stage. The fungal abundance decreased gradually with stand age, with the lowest fungal diversity at the near-mature stand stage. During the whole growth process, the bacterial community was more limited by soil pH, nitrogen, and phosphorus. After introducing *P. bournei* into a Chinese fir plantation, the abundance and diversity of the bacterial community did not improve, and the abundance of the fungal community did not increase. However, soil nutrients, pH, and fungal community diversity were significantly improved. The results of these studies indicate that the introduction of broad-leaved tree species not only increased soil nutrient content, but also had a significant effect on the increase in the diversity of soil fungal communities, making the microbial communities of mixed forests more diverse.

**Keywords:** Chinese fir; forest age and stand structure; soil nutrient; soil microbial community

## 1. Introduction

Chinese fir (*Cunninghamia lanceolata*) is a fast-growing coniferous tree that has been widely cultivated for timber production in southern China [1]. The species makes an important contribution to environmental protection, the provision of substantial timber products, and ecological construction. Due to the rising demand for timber, Chinese fir

plantations usually grow in a monoculture pattern accompanied by a continuous crop rotation [2]. These practices decrease soil fertility and ecosystem productivity because of the excessive afforestation density [3]. For example, Zhang Hong et al. [4] found that the soil phosphorus content gradually decreased with the increase in planting generations. An in-depth understanding of the reasons for declines in soil quality in C. lanceolata plantations is extremely important for the sustainable management of these plantations in China.

Governments and forest managers have been investigating forest management practices that can improve ecosystem functions and services [5]. The replacement of monocultures with mixed plantations is an effective afforestation method to mitigate soil erosion and enhance forest productivity [6–8]. Recently, several native broad-leaved trees were introduced into monoculture *C. lanceolata* plantations to improve soil status [8]. Phoebe bournei is an economically important, valuable, and rare broad-leaved native tree, and is required in mixed forests [1,9]. A mixed forest can increase the diversity of plants, which in turn positively affects microbial activity and influences soil microorganisms in a number of ways [10]. It affects soil microbes in several ways, for example, by accelerating litter decomposition and changing root exudates [11–13]. Not only are soil microorganisms an important driving factor for plants to obtain available nutrients for growth [14,15], they are also very sensitive to changes in soil properties [15,16].

The conversion of stand types may impact the structure of soil microbial communities by influencing soil environment [17,18]. In addition, soil microorganisms can influence forest productivity by participating in nutrient cycling [19] and soil organic matter turnover [20], while ensuring the reproduction of microbial communities. Therefore, it is necessary to understand the relationship and interaction between tree species characteristics, soil nutrient changes, and microbial community structure characteristics for the sustainable management of coniferous and broad-leaved mixed forests [21,22].

Furthermore, bacteria and fungi play different and important roles in biogeochemical cycles [23], because the structure and diversity of microbial communities may respond differently to soils in different ecosystems [24]. For example, fungi mainly decompose refractory cellulose and lignin [25,26]. Bacteria are mainly responsible for nutrient cycling [27]. However, poor knowledge exists about the differences in the responses of bacterial and fungal communities after the transformation of Chinese fir plantations into mixed forests and their relationship with soil characteristics.

Current studies in Chinese fir plantations mainly focus on soil properties, differences in soil properties [28], nutrient cycling rates [29], and microbial structure [30] between *C. lanceolata* plantation forests under monoculture and mixed plantation patterns. Wang et al. found that soil nutrients and microorganisms were enhanced after constructing a mixed forest [31]. Lei et al. found that soil microbial diversity was significantly higher in a near-natural forest versus in a *C. lanceolate* plantation [32]. Chu et al. reported that diversity and abundance of soil microbial communities in natural secondary forests was significantly higher than that in plantations [33].

Moreover, the significant effect of age on soil properties is widely recognized. However, there are few reports on the effects of stand-type changes in soil characteristics, as well as few studies that further research impacts on the diversity, composition, and abundance of microbial communities, determining the key driving factors (soil chemical factors and enzyme activities) affecting soil microbial communities in the single and mixed cultivation of Chinese fir (*C. lanceolata* × *P. bournei*). Moreover, there are relatively few studies on the relationship between microbial community structure and composition, as well as soil properties and enzymes. This greatly limits us in understanding the reaction of microbial communities to forest age and stand type. Therefore, an in-depth understanding of the effects of stand age and stand type on microbial communities will help us to make more rational plantation management methods.

Here, we aimed to fill the knowledge gaps in the effect of stand age and stand type on soil fertility and microbial diversity, composition, and structure in Chinese fir plantations.

We also aimed to advance our understanding of the influence of stand age on microbial structure and composition and explore key limiting factors that affect the soil microbial community structure. Therefore, our specific objectives were to determine: (1) how soil nutrients and enzyme activities vary with stand age and stand type; (2) how microbial diversity and composition changes with stand age and stand type; and (3) the key factors that affect the structure of soil microbial communities in plantation forests.

## 2. Material and Methods

### 2.1. Description of the Field Trial and Soil Sample Collection

The research site is located in Guanshan forest farm, Yongfeng County, Ji'an City, Jiangxi Province (115°17′–115°56′ E, 26°38′–27°32′ N). The climate is humid and is in the mid subtropical zone, with abundant rainfall and sufficient light. The average annual temperature is 18 °C and the average annual rainfall is 1627.3 mm. Red soil is a typical local soil type with the largest area. The main planting species are *C. lanceolata*, Slash pine, Moso bamboo, *P. bournei*, and Liquidambar formosana. Three main sample plots adjacent to Guanshan forest farm are selected: Dazhou Mountain Branch (115°29′7″ E, 27°14′48″ N), Dongmaokeng branch (115°33′10″ E, 27°10′43″ N), and Yuyuan branch (115°34′39″ E, 27°14′13″ N).

In December 2017, we sampled *C. lanceolata* plantations of over-ripe forest (OR), near-mature forest (NM), and young forest (YF), which had the same soil type, site conditions, and disturbance history. We also sampled a mixed forest of *C. lanceolata* and *P. bournei* with two forest ages, one near-mature *C. lanceolata* forest mixed with 5-year-old *P. bournei* (NM + PB05), and a near-mature *C. lanceolata* forest mixed with 10-year-old *P. bournei* (NM + PB10). Each sample plot was divided into three 20 m × 20 m quadrants, and we selected three trees randomly in each quadrant. The plant and animal residues were removed from the soil surface within 30 cm from the stem of *C. lanceolata*, and the soil profile was dug to a depth of 0–30 cm. The roots were taken near the main root system with fine roots, put into sterile self-sealing bags, and shaken vigorously for 1 min to separate the roots from the soil. We collected total 45 soil samples, which was used as rhizosphere soil, and were stored in a low-temperature refrigerator and transported to the laboratory for analysis.

### 2.2. Soil Properties and Soil Enzyme Activities Analysis

The samples used for microbial sequencing analysis were stored in a −70 °C ultra-low temperature refrigerator. The soil samples used for the remaining test indicators were placed in an oven, dried at a temperature of 105° to a constant weight, and then filtered using a 2 mm mesh sieve to remove impurities such as plant roots and stones.

All soil indicators are in accordance with the standard agreement. The glass electrode of PHSJ-3D acidity meter produced by Shanghai Rex Company was suspended in a 1 mol/L KCL solution (*w*: *v*, 1: 5) to determine the pH value of rhizosphere soil. Total carbon and total nitrogen were determined using an elemental analyzer (Vario EL III, Elementar, Nakano, Tokyo). SOC was measured using an elemental analyzer after removing carbonates from air-dried samples acidified with dilute hydrochloric acid (5%). Soil total phosphorus: acid soluble-molybdenum antimony anti-colorimetric determination. Soil available phosphorus: hydrochloric acid–ammonium fluoride extraction method. Soils' total potassium: Hydrofluoric acid–perchloric acid digestion method. Soils' available potassium: Ammonium acetate extraction method. Soils' available nitrogen: Petri dish diffusion method [34–38].

All enzyme activities were determined in accordance with the standard agreement [39]. Soil sucrase: 3,5-dinitrosalicylic acid colorimetric method was used to determine the quality of glucose produced by 1 g of soil cultured at 37 °C for 24 h to characterize the activity of invertase. Urease: Using a sodium salicylate-sodium dichloroisocyanurate colorimetric method, 1 g of soil was cultured at 37 °C for 2 h. Acid phosphatase: Measured by p-nitrophenyl disodium phosphate colorimetric method, 1 g soil was cultured at 37 °C for 1 h.

*2.3. DNA Extraction, PCR Amplification, and Illumina MiSeq Sequencing of Soil Microorganisms*

According to the manufacturer 's instructions, PowerSoil DNA extraction kit (Mo Bio Laboratories Inc., Carlsbad, CA, USA) was used for rhizosphere soil DNA extraction. The purity of the sample DNA was detected by a NanoPhotometer spectrophotometer, and the concentration was detected by Qubit2.0 fluorometer. The V3–V4 region of the 16S rDNA gene of rhizosphere soil bacteria were amplified with primers 341F (5′-CCTACGG NGGCWGCAG-3′) and 805R (5′-GACTACHVGGGT ATCTAATCC-3′), while the rhizosphere soil fungi were amplified with primers ITS1 (5′-ACC TGCGGARGGATCA-3′) and B58S3 (5′-GATCCRTTGYTRAAAGTT-3′). TaKaRa EXtaq enzyme was used during the amplification process to ensure the success and accuracy of the amplification, and each sample was subjected to three PCR reactions. After preliminary quantification using Qubit2.0, the DNA was diluted to 1 ng/uL with sterile water. All PCR reactions were using PhusionÂ® High-Fidelity PCR Master Mix (New England BioLabs, Beverly, MA, USA).

Firstly, the mixed PCR products with equal density ratio were purified by GeneJET gel extraction kit (Thermo Scientific, Waltham, MA, USA). Then, NEB Next Ultra DNA Library Prep Kit for Illumina (NEB, Ipswich, MA, USA) was used to generate sequencing libraries and index codes, and Qubit @ 2.0 fluorescence analyzer (Life Technologies, Carlsbad, CA, USA) and Agilent Bioanalyzer 2100 system were used for the quality assessment. Finally, the library was sequenced using the 250 pair-end protocol of the Illumina HiSeq2500 instrument.

In order to obtain a credible target sequence and facilitate the subsequent analysis, the sequence obtained by sequencing was first filtered and removed for low-quality base, Ns, and linker contamination sequences. Then, the PEAR method [40] was used to splice the corresponding sequence fragments of the paired-end sequencing, and the quality control process of the spliced sequence was removed according to QIIME 1.8.0 to remove the quality filtering of the original label and remove 'chi-mera'. Finally, the sequences with 97% similarity as the threshold were assigned to the OTUs, and the representative sequences of each OTU were classified using the RDP classifier or Closed Reference against the Greengenes 16S rRNA reference database [41–43]. The work was provided by Annoroad Gene Technology Co., Ltd. Beijing, China. Finally, we obtained 1,023,685 bacterial sequences and 477,975 fungal sequences from all rhizosphere soil samples.

*2.4. Statistical Analysis*

One-way analysis of variance in SPSS software (IBM SPSS Statistics 25.0; IBM Corp., Armonk, NY, USA) was used to test the differences in soil properties and enzyme activities, as well as diversity (Shannon, Simpson) and abundance (OTU, Chao1) among different communities. The Wayne diagram was drawn using InteractiVenn (http://www.interactivenn.net (accessed on 13 March 2023)) to show the common and unique OTUs of each community [44]. Linear discriminant analysis (LDA) and effect size (LEfSe; http://huttenhower.sph.harvard.edu/galaxy/root?tool_id=PICRUSt_normaliz (accessed on 13 March 2023)) was used to identify differences in bacterial/fungal taxa among communities [45]. The PCoA analysis in Canoco 5.0 software was used to show the differences in microbial population structure by the relative abundance of OTU. A fungal and bacterial Mantel test analysis using Spearman and Pearson functions of the Rstudio Vegan package revealed factors affecting the abundance of soil microbial communities in fir plantations and mixed forests.

## 3. Results

### 3.1. Soil Chemical Properties and Enzyme Activities

We found that SOC, TK, and AK contents gradually increased with the stand age of *C. lanceolata* plantations, with significant differences in TOC between the young forest and near-mature forest, non-significant differences in the SOC contents among the developmental stages, significant differences in TK contents among developmental stages, and significant differences in AK contents between the young forest and near-mature forest. pH gradually

decreased with increasing stand age, with significant differences among the developmental stages (Table 1). TN, AN, TP, and AP contents showed a "V"-shaped trend with the increasing age of fir stands, with significant differences between NM and OR for TN and AN for all developmental stages, and TP and AP for NM and the rest of the stages. S-SC activity increased gradually with increasing stand age, but there were no significant differences among all developmental stages ($p < 0.05$, Table 1).

The soil TOC and TN contents differed significantly between NM and NM + PB10 in the Chinese fir forest and mixed stands of fir trees at the same developmental stage, but SOC content was not significantly different. Soil AN, and TP contents were significantly lower in the Chinese fir forest than in mixed forests. Soil AP content was significantly lower than that of NM + PB05 and higher than that of NM + PB10. Soil TK and AK content and pH were significantly lower in the Chinese fir forest than in mixed forests. The ACP activity was significantly lower than that of NM + PB05 and not significantly different from that of NM + PB10. S-SC was significantly lower in the Chinese fir forest than in mixed forests. SUE was significantly higher in the Chinese fir forest than in mixed forests ($p < 0.05$, Table 1).

*3.2. Characterization of Diversity of Soil Microbial Community*

The bacterial community richness and diversity (Table 2) showed a significant difference in differently aged *C. lanceolata* forests. OTU, ACE, and chao1 showed an overall upward trend in growth and then decline. Bacterial abundance mainly reached the highest in the near-mature forest stage, but both the Shannon and Simpson indices showed no difference (Table 2, $p < 0.05$).

In the *C. lanceolata* plantation (near-mature forest) and *C. lanceolata*–*P. bournei* mixed forest of the same stand age, we found that plantations were higher in bacterial diversity than mixed forests. The OTU, ACE, and chao1 indices showed a tendency to increase with the increasing stand age in the NM + PB05 forest and NM + PB10 forest, but there was no significant difference (Table 2, $p < 0.05$ for all pairs).

Shannon and Simpson indices showed a decreasing and then increasing trend with stand age, and a significant difference was found between the younger and near-mature stages. In mixed forests, the Shannon and Simpson indices did not vary significantly. The diversity indices of fir plantations at the same developmental stage were lower than those of mixed forests, and the differences in indices were significant (Table 3, $p < 0.05$ for all pairs).

For the fungal community, community richness was significantly different in differently aged *C. lanceolata* forests. Chao1 index values gradually decreased with the increasing stand age and reached the lowest value at the over-ripe stand stage. In the mixed forest, it had an upward trend of gradually increasing in abundance with the increasing forest age (Table 3, $p < 0.05$ for all pairs).

**Table 1.** Soil physicochemical characteristics and enzyme activities in the different ages of Chinese fir forest and mixed forests.

| | TOC (g/kg) | SOC (mg/g) | TN (g/kg) | AN (mg/g) | TP (g/kg) | AP (mg/g) | TK (g/kg) | AK (mg/g) | pH | ACP ($mg \cdot g^{-1} \cdot h^{-1}$) | S-SC ($mg \cdot g^{-1} \cdot h^{-1}$) | SUE ($mg \cdot g^{-1} \cdot h^{-1}$) |
|---|---|---|---|---|---|---|---|---|---|---|---|---|
| YF | 9.06 ± 0.79 a | 0.94 ± 0.51 a | 1.29 ± 0.10 bc | 60.76 ± 8.08 bc | 0.22 ± 0.56 b | 0.667 ± 0.09 b | 7.47 ± 0.50 a | 22.69 ± 4.10 a | 4.67 ± 0.20 c | 0.46 ± 0.01 a | 3.27 ± 0.52 a | 28.38 ± 8.50 b |
| NM | 14 ± 0.72 b | 1.14 ± 0.87 ab | 0.68 ± 0.03 a | 33.32 ± 5.27 a | 0.07 ± 0.00 a | 0.37 ± 0.10 b | 12.68 ± 0.25 b | 33.41 ± 3.37 b | 4.19 ± 0.71 b | 0.52 ± 0.02 ab | 4.72 ± 0.80 a | 58.11 ± 9.60 d |
| OR | 14.7 ± 0.45 bc | 1.21 ± 0.05 ab | 1.13 ± 0.15 b | 56.04 ± 6.99 b | 0.264 ± 0.34 b | 0.56 ± 0.05 ab | 15.5 ± 0.95 c | 36.62 ± 6.70 b | 3.85 ± 0.07 a | 0.50 ± 0.01 a | 5.10 ± 0.37 a | 41.63 ± 0.09 c |
| NM + PB05 | 14.7 ± 1.4 bc | 1.18 ± 0.29 ab | 0.75 ± 0.15 a | 72.38 ± 10.72 c | 0.3719 ± 0.00 c | 4.04 ± 0.13 d | 14.9 ± 1.45 c | 49.32 ± 0.51 c | 4.62 ± 0.08 c | 0.70 ± 0.11 c | 11.81 ± 3.53 b | 13.18 ± 1.65 a |
| NM + PB10 | 16.1 ± 1.3 c | 1.34 ± 0.76 b | 1.35 ± 0.05 c | 51.36 ± 10.59 b | 0.226 ± 0.23 b | 2.73 ± 0.20 c | 15.13 ± 0.86 c | 34.23 ± 5.89 b | 4.63 ± 0.13 c | 0.56 ± 0.10 bc | 14.14 ± 1.23 b | 16.73 ± 1.79 a |

Note: One-way ANOVA was used for analysis, $p < 0.05$. Different letters represent significant differences between groups. Data: mean ± SD ($n = 3$). TOC: total organic carbon; SOC: soil organic carbon; TN: total nitrogen; AN: available nitrogen; TP: total phosphorus; AP: available phosphorus; TK: total potassium; AK: available potassium; ACP: acid phosphatase; S-SC: sucrose; SUE: urease; YF: Young forest; NM: Near-mature forest; OR: Overripe forest.; NM + PB05: Near-mature *C. lanceolata* plantation mixed with 5 years old *P. bournei*; NM + PB10: Near-mature *C. lanceolata* plantation mixed with 10-year-old *P. bournei*.

**Table 2.** The bacterial diversity in different ages of Chinese fir plantations and mixed forests.

| Treatment | OTU | ACE | Chao1 | Shannon | Simpson |
|---|---|---|---|---|---|
| YF | 1671.67 ± 89.07 b | 1988.67 ± 60.43 c | 2010.67 ± 71.93 c | 5.81 ± 0.23 a | 0.88 ± 0.40 a |
| NM | 1944.67 ± 131.80 a | 2319.67 ± 111.79 a | 2342.00 ± 112.93 a | 6.02 ± 0.12 a | 0.64 ± 0.02 a |
| OR | 1761.00 ± 69.46 ab | 2071.00 ± 125.41 bc | 2101.67 ± 155.00 bc | 6.02 ± 0.17 a | 0.62 ± 0.13 a |
| NM + PB05 | 1831.00 ± 100.54 ab | 2172.33 ± 155.99 abc | 2210.33 ± 144.67 abc | 5.95 ± 0.17 a | 0.67 ± 0.17 a |
| NM + PB10 | 1889.33 ± 165.24 ab | 2279.67 ± 140.63 ab | 2311.00 ± 93.21 ab | 5.97 ± 0.15 a | 0.67 ± 0.08 a |

Note: YF: Young forest; NM: Near-mature forest; OR: Overripe forest.; NM + PB05: Near-mature *C. lanceolata* plantation mixed with 5 years old *P. bournei*; NM + PB10: Near-mature *C. lanceolata* plantation mixed with 10-year-old *P. bournei*. Different letters represent significant differences in Chinese fir plantations and mixed forests of different ages ($p < 0.05$).

**Table 3.** The fungal diversity in different ages of Chinese fir plantations and mixed forests.

| Treatment | Chao1 | Observed Species | PD Whole Tree | Shannon | Simpson |
|---|---|---|---|---|---|
| YF | 779.93 ± 93.18 ab | 676.00 ± 96.02 a | 173.13 ± 20.75 a | 6.40 ± 0.60 a | 0.97 ± 0.02 a |
| NM | 681.17 ± 80.70 ab | 549.00 ± 131.53 a | 157.88 ± 23.00 a | 3.34 ± 1.95 b | 0.57 ± 0.33 b |
| OR | 584.43 ± 217.54 b | 487.33 ± 202.21 a | 145.97 ± 51.81 a | 3.93 ± 2.32 ab | 0.67 ± 0.33 ab |
| NM + PB05 | 768.49 ± 33.36 ab | 697.67 ± 47.60 a | 193.93 ± 8.98 a | 6.23 ± 0.71 a | 0.94 ± 0.05 ab |
| NM + PB10 | 826.32 ± 130.59 a | 687.00 ± 150.12 a | 189.86 ± 29.42 a | 5.10 ± 1.31 ab | 0.87 ± 0.13 ab |

Note: YF: Young forest; NM: Near-mature forest; OR: Overripe forest.; NM + PB05: Near-mature *C. lanceolata* plantation mixed with 5 years old *P. bournei*; NM + PB10: Near-mature *C. lanceolata* plantation mixed with 10-year-old *P. bournei.* Different letters represent significant differences in Chinese fir plantations and mixed forests of different ages ($p < 0.05$).

### 3.3. Differences in the Taxonomic Level Characteristics of Soil Microbial Communities

All sequences were assigned to 48,980 OTUs across all samples and categorized to the bacterial domain, which includes 27 bacterial phyla, 70 classes, 143 orders, 243 families, and 375 genera. The most common phylum in OTUs was *Proteobacteria*, accounting for 32.87%, followed by *Acidobacteria* (OTUs: 28.24%), *Chloroflexi* (OTUs: 15.02%), *Actinobacteria* (OTUs: 9.55%), *Planctomycetes* (OTUs: 4.09%), *Verrucomicrobia* (OTUs: 3.03%), and *Firmicutes* (OTUs: 2.39%). A detailed abundance of phyla was shown in Figure 1.

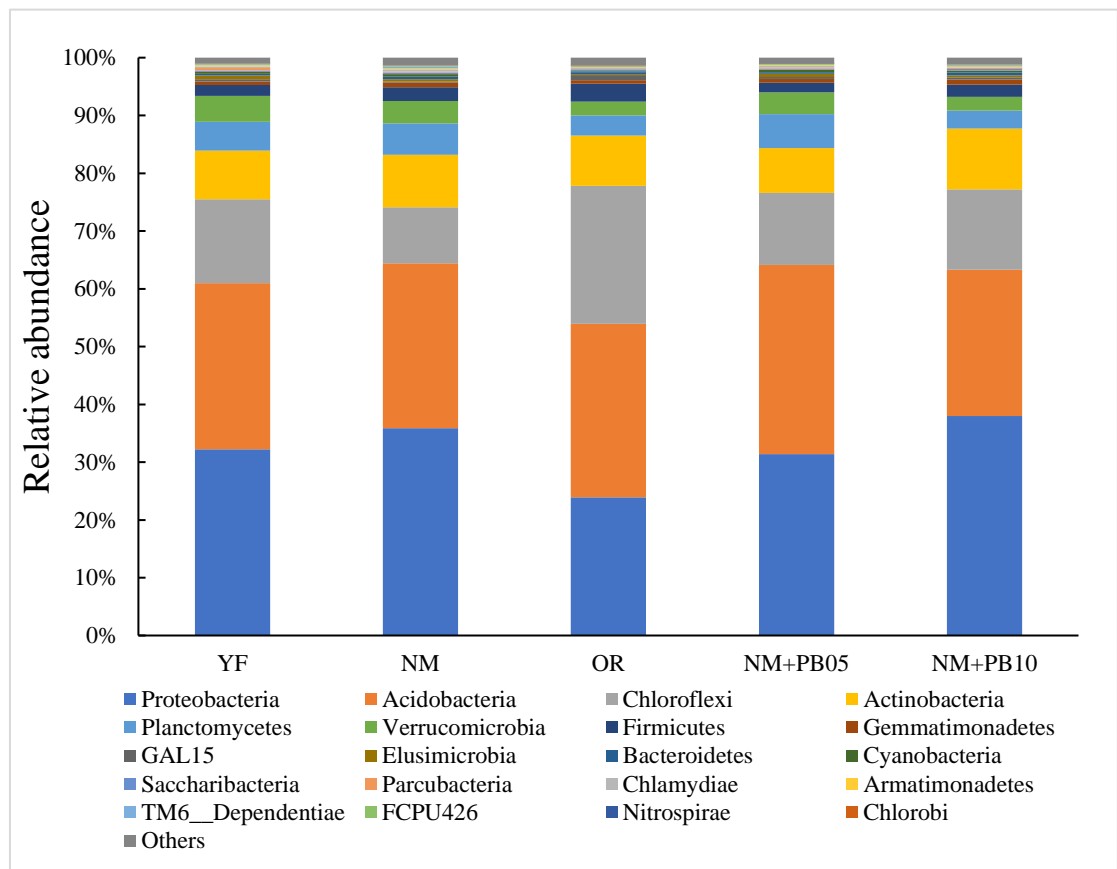

**Figure 1.** The relative abundance of bacterial phylum in young (YF), near-mature (NM), and over-ripe (OR) Chinese fir plantation; near-mature Chinese fir plantation mixed with 5-year-old *P. bournei* (NM + PB05); Chinese fir plantation mixed with 10-year-old *P. bournei* (NM + PB10).

Lefse analysis showed differences in the abundance of certain taxa in soil samples (Figure 2). When comparing microbial communities in pure stands of fir trees of different ages, we found that the order *Sp. Fcpu453* and the class *Betaproteobacteria* were the dominant species of the YF stands. In the communities of the OR stands, the family *Sp. HSB_ OF53_ F07* and the family *Sp. KF_ JG30_ B3* were the dominant species, which were significantly higher than other microbial communities. The bacterial community of the NM forest had no dominant species (Figure 2a). By comparing the microbial communities of fir plantations of different ages, we found that the order *Acidimicrobiales*, the family *Gemmatimonadaceae*, and the phylum *Gemmatimonadetes* were the dominant species in the NM + PB10 mixed forest communities. In the NM forest community, the class *Chlamydiae*, the phylum *Chlamydiae*, and the order *Chlamydiales* were the main dominant species affecting the soil bacterial community. In contrast, there was no high abundance of bacteria in the mixed forest community (Figure 2b).

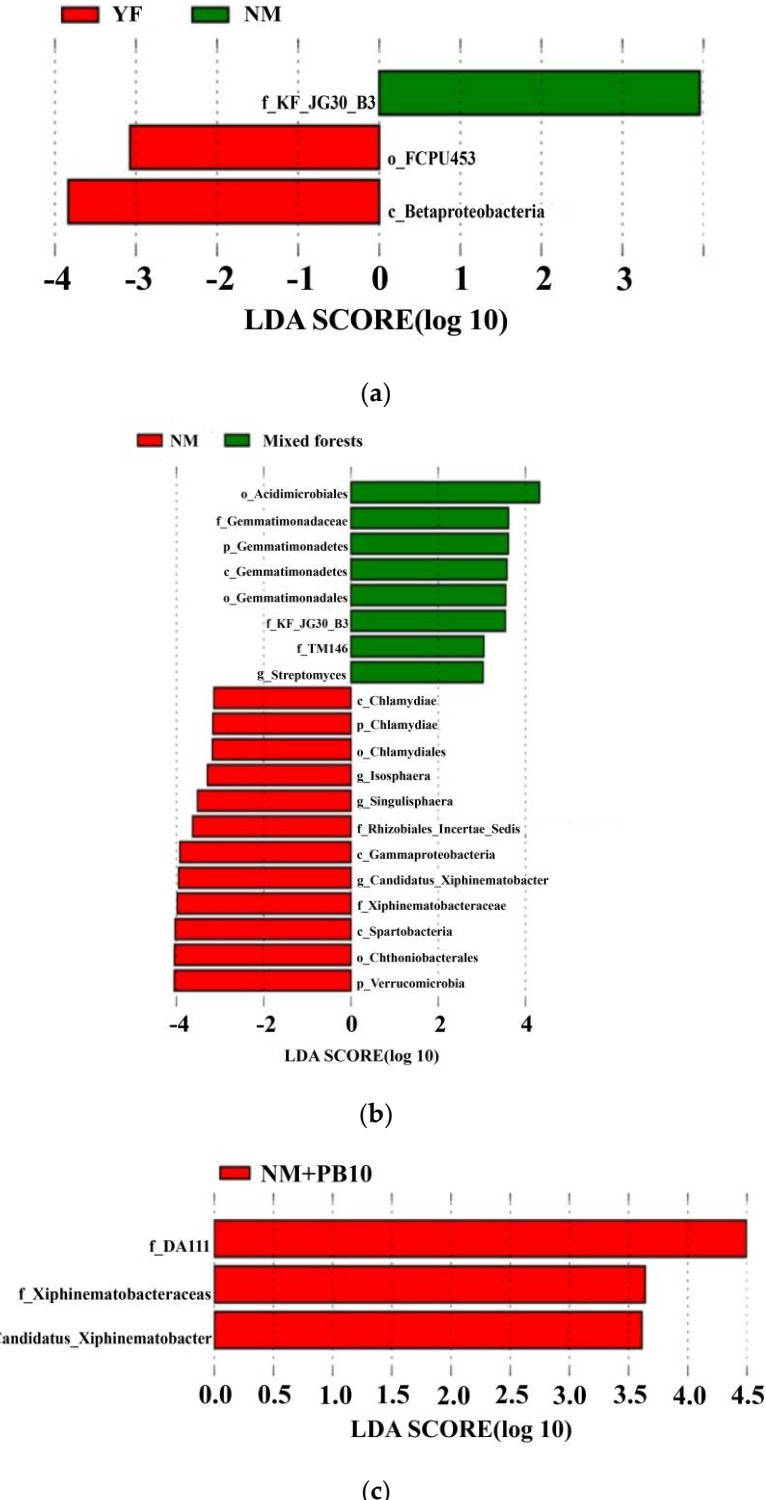

**Figure 2.** Lefse analysis of bacterial taxonomic levels in *C. lanceolata* plantations and Chinese fir-*P. bournei* mixed forests. (**a**) Lefse analysis of bacterial taxonomic levels in *C. lanceolata* plantations of different forest ages; (**b**) Lefse analysis of bacterial taxonomic levels in *C. lanceolata* plantations and Chinese fir-*P. bournei* mixed forests at the same developmental stage; and (**c**) Lefse analysis of bacterial taxonomic levels in Chinese fir–*P. bournei* mixed forests of different forest ages. Abbreviations: p: phylum, c: class, o: order, f: family, and g: genus. LDA > 3.5, *p* < 0.05.

Lefse analysis showed that the abundance of certain taxa differed in samples from mixed forests of different ages. In the NM + PB10 community, the family *DA111*, the family

*Xiphinematobacteraceae*, and the genus *Candidatus_Xiphinematobacter* were the dominant species. In contrast, there was no high abundance of bacteria in the NM + PB05 community (Figure 2c).

In the fungi community (Figure 3), all sequences were classified to the fungi domain and assigned to 31,865 OTUs across all samples, including 11 fungi phyla, 19 classes, 20 orders, 19 families, and 20 genera. The phylum with the most OTUs was *Ascomycota* (54.12%), followed by *Basidiomycota* (10.81%) and *Mortierellomycota* (2.34%).

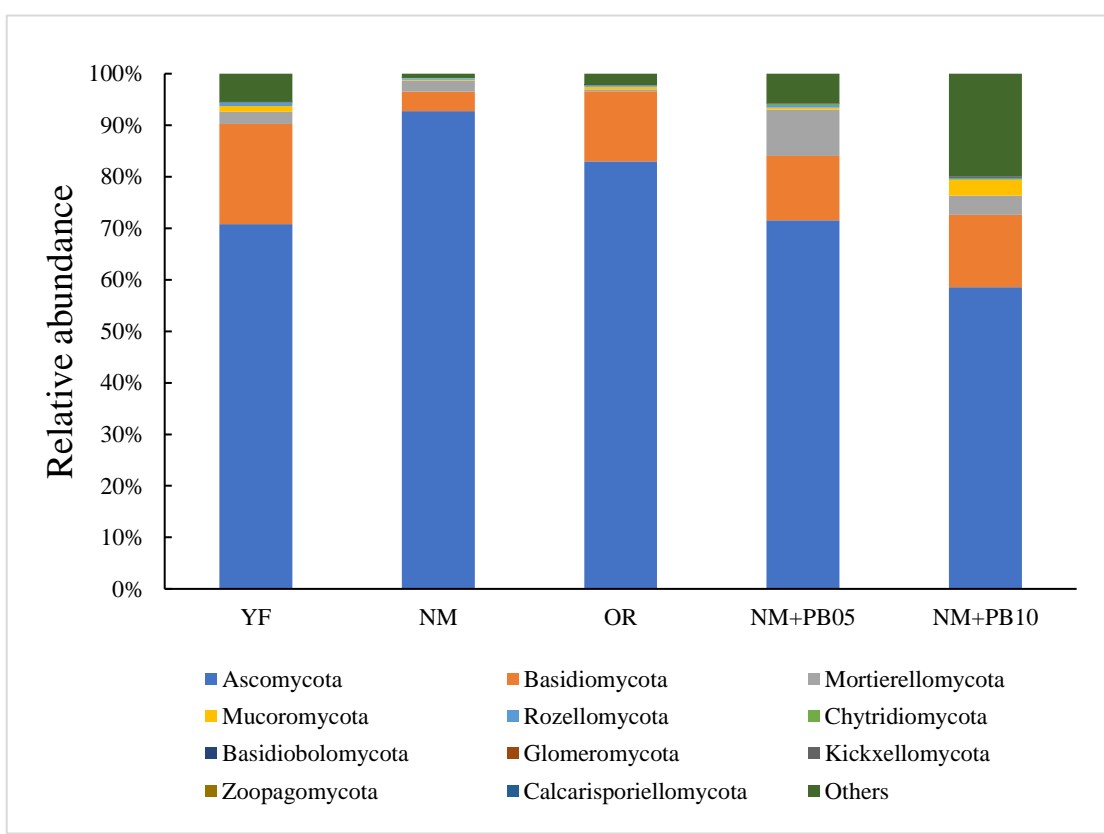

**Figure 3.** The relative abundance of fungi phylum in young (YF), near-mature (NM), and over-ripe (OR) Chinese fir plantations; near-mature Chinese fir plantation mixed with 5-year-old *P. bournei* (NM + PB05); Chinese fir plantation mixed with 10-year-old *P. bournei* (NM + PB10).

Lefse analysis showed differences in great numbers of dominant taxa in the soil samples. The abundance of fungal taxa in fir plantation forest samples differed between stand age and stand type. The genus *Ceratobasidium* and the genus *Leptodontidium* were the dominant species in the NM community. The order *Sordariales*, the order *Chaetothyriales*, and the order *Agaricales* were dominant in the YF community. There were no dominant species in the OR community (Figure 4a). The phylum *Basidiomycota*, the class *Cystobasidiomycetes*, the genus *Bannoa*, and the order *Xylariales* were the dominant species in the mixed forests community. In the NM community, the family *Clavulinaceae*, the family *Cordycipitaceae*, and the class *Archaeorhizomycetes* were the main species affecting the community (Figure 4b).

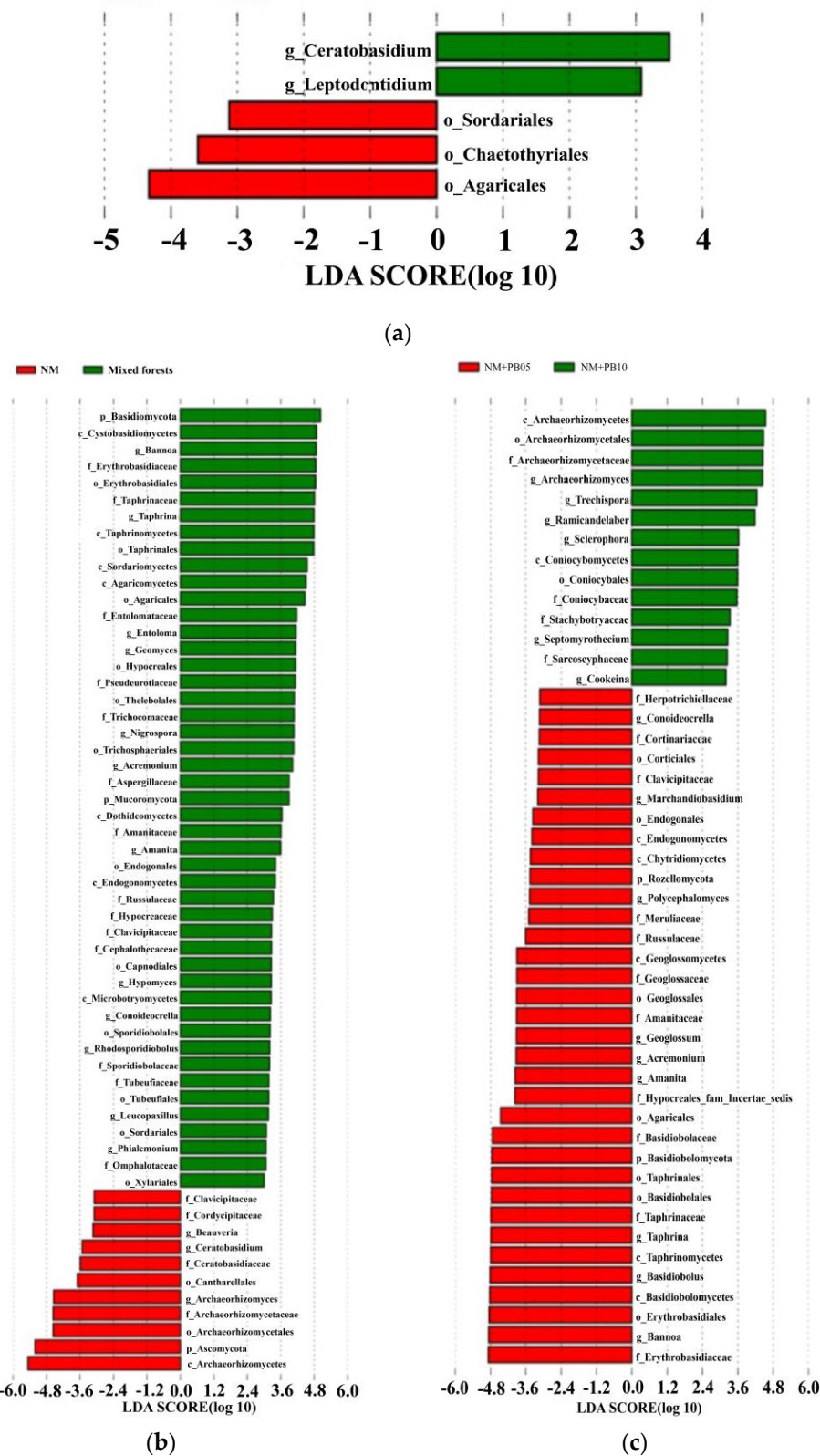

**Figure 4.** Lefse analysis of fungal taxonomic levels in pure and mixed cedar-heath forests. (**a**) Lefse analysis of fungal taxonomic levels in *C. lanceolata* plantations of different ages; (**b**) Lefse analysis of fungal taxonomic levels in plantations and mixed forests at the same developmental stage; (**c**) Lefse analysis of fungal taxonomic levels in mixed *C. lanceolate–P. bournei* forests of different ages. Abbreviations: p: phylum, c: class, o: order, f: family, and g: genus. LDA > 3.5, *p* < 0.05.

Lefse analysis showed that the abundance of certain taxa differed in samples from mixed forests of different forest ages. We found that the class *Archaeorhizomycetes*, the order *Archaeorhizomycetales*, and the family *Erythrobasidiaceae*, as well as other bacteria, were more abundant in the NM + PB05 mixed forest community. The family *Herpotrichiellaceae*, the genus *Conoideocrella*, the family *Corticiales*, and other bacteria were more abundant in the NM + PB10 mixed forest community (Figure 4c).

### 3.4. Structure of Microbial Community at OUT Level

In all treated samples, there were 12,503 OUT sequences in the bacterial community, and the bacterial community shared by all communities accounted for 11.4% of the OUT. The bacterial community of the Chinese fir plantation at different ages accounted for 47% of the total. The bacterial community of the mixed forest accounted for 53% of the OTUs (Figure 5a).

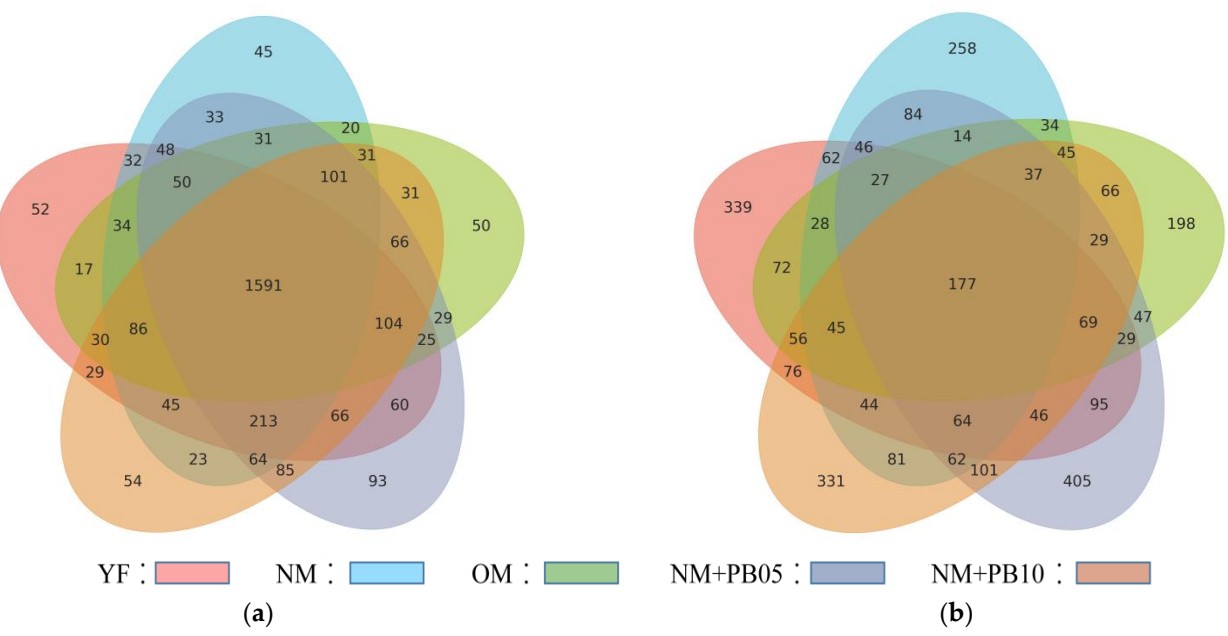

**Figure 5.** Venn plot shows the unique and shared OTUs among samples from Chinese fir plantation and mixed forest of different forest ages in the bacterial (**a**) and fungal (**b**) communities.

There were 6017 OTU sequences in all samples of fungal communities, of which all fungal communities had 3% of the same OTU. The fungal community of the *C. lanceolata* plantation at different ages accounted for 45% of the total communities, and the fungal community in the mixed forest accounted for 55% of the total communities (Figure 5b).

The first and second axes of the PCoA analysis based on the OTUs' data of differently aged Chinese fir plantations' soil bacterial communities explained 20.08% and 39.42% of the variance, respectively, suggesting that soil bacterial communities differed along the stand age gradient (Figure 6a). The PCoA analysis Axis1 explained 27.39% of the variance and the Axis2 explained 18.45% of the variance, illustrating that there were differences in the rhizosphere soil bacterial communities between Chinese fir plantations and Chinese fir–*P. bournei* mixed forests (Figure 6b).

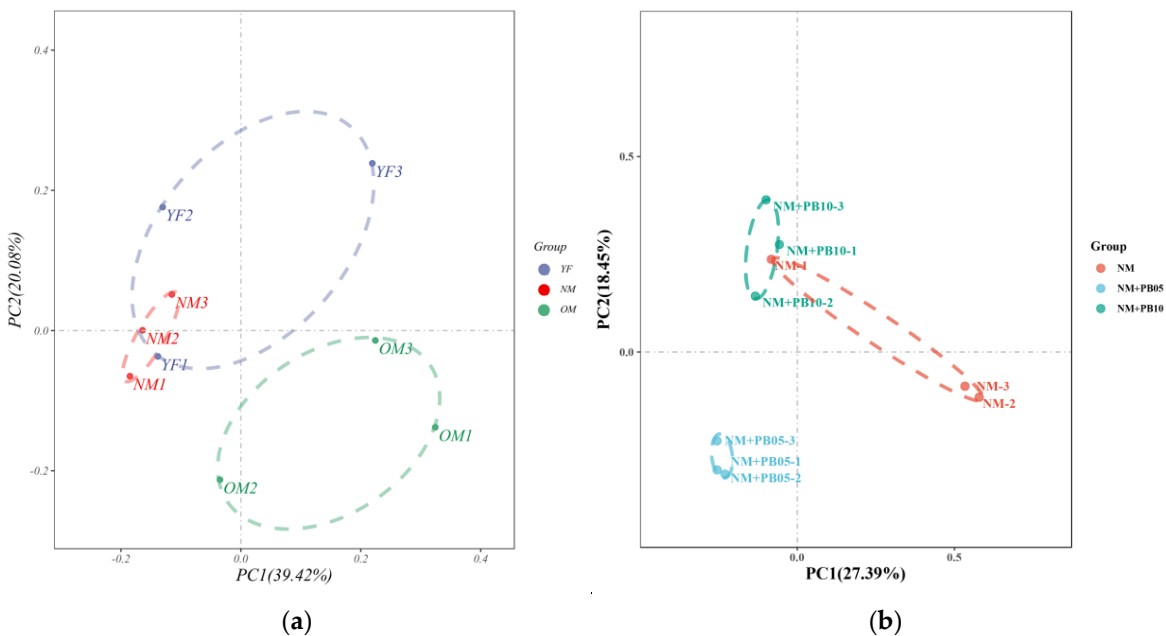

**Figure 6.** The PCoA results explained 59.47% of observed variation in rhizosphere soil bacteria communities in Chinese fir plantations of different ages (**a**) and 45.84% of observed variation in rhizosphere soil bacterial communities of Chinese fir–*P. bournei* mixed forests at the same developmental stage (**b**). The color of the points represents different soil bacterial communities (PerMANOVA test, $p < 0.05$).

Based on the OTUs data from differently aged Chinese fir plantations' rhizosphere soil fungal communities, the first and second PCoA axes explained 34.04% and 19.94%, respectively, of the variance. Thus, soil fungal communities differed along a gradient of forest age (Figure 7a). In a PCoA analysis of rhizosphere soil fungal communities in Chinese fir plantations and Chinese fir–*P. bournei* mixed forests at the same developmental stage, the first axis explained 32.36% of the variation and the second axis explained 23.09% of the variation, respectively (Figure 7b).

### 3.5. Identification of Key Drivers Affecting Microbial Community Structure

Soil microbial community is very sensitive to changes in soil factors, the age of Chinese fir plantation forests, and the construction of mixed forests. Soil properties continually change, and the microbial community's feedback to its factors differ, resulting in changes in the abundance of the microbial community (Figure 6). In the young forest stage, the soil bacterial community abundance was strongly influenced by soil pH ($r = 0.99$ **, Figure 8A). TK and pH had the greatest influence on the soil bacterial community abundance in the near-mature forest stage ($r = 0.98$ **, Figure 8B), and TP and ACP had the greatest influence in the over-ripe forest stage ($r = 0.99$ **, Figure 8C). AN dominated the soil bacterial community abundance in the mixed forest ($r = 0.99$ **, Figure 8D).

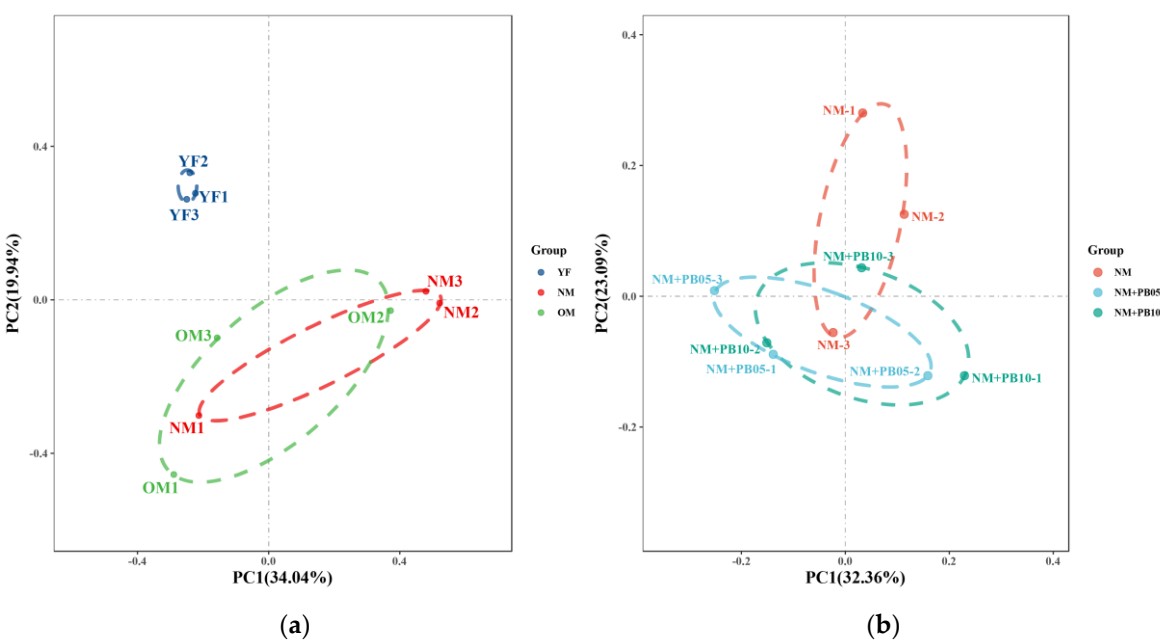

**Figure 7.** According to the results of PcoA, there is 53.98% of observed variation in rhizosphere soil fungal communities of differently aged Chinese fir plantations (**a**) and 55.45% of observed variation of Chinese fir–P. bournei mixed forests at the same developmental stage (**b**). The color of the dots represents the different soil bacterial communities.

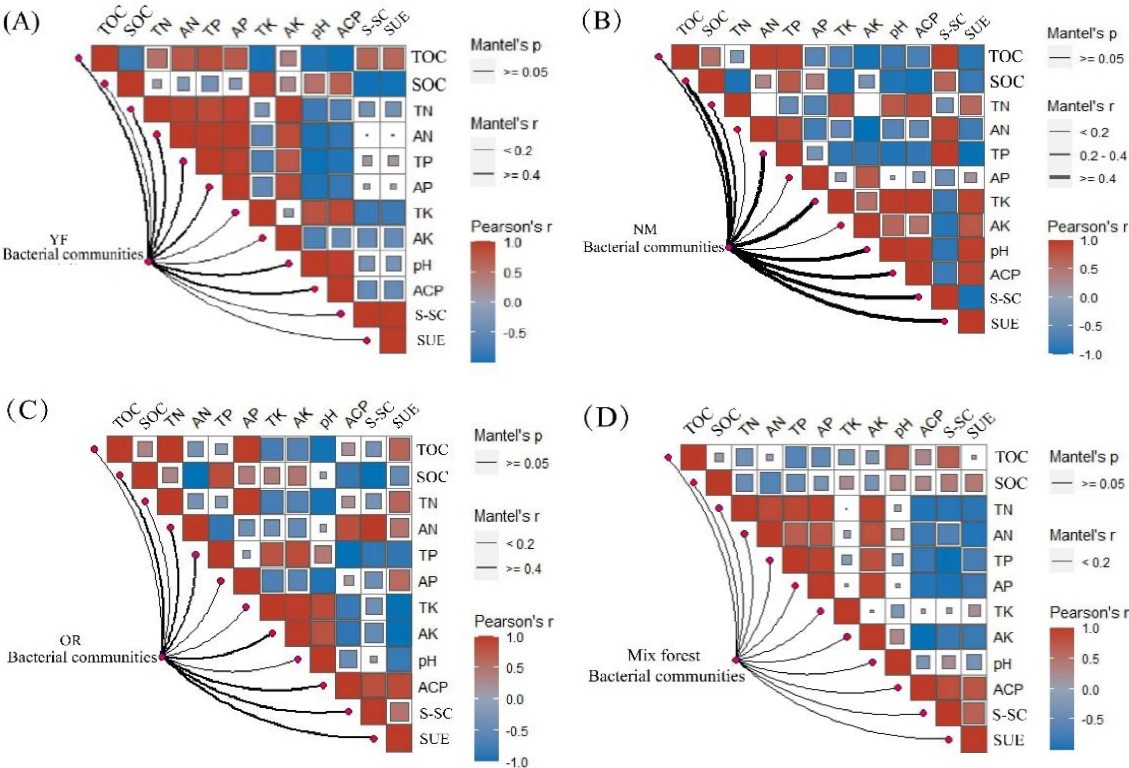

**Figure 8.** Identification of key driving factors affecting bacterial community structure in different forest ages and stand types. (**A**) Correlation of soil bacterial communities with soil nutrients and enzyme activities at the young forest; (**B**) Correlation of soil bacterial communities with soil nutrients and enzyme activities in near-mature forests; (**C**) Correlation of soil bacterial communities with soil

nutrients and enzyme activities in overripe forests; (**D**) Correlation of soil bacterial communities with soil nutrients and enzyme activities in Chinese fir–*P. bournei* mixed forests. Note: Significant correlation at Mantel's r ≥ 0.4. Correlation is significant at 0.2 ≤ Mantel's r ≤ 0.4. Data: means ± SD (*n* = 3). Color gradient indicates Spearman's correlation coefficient; the side width corresponds to the correlation coefficient for distance calculated by Mantel; color shade indicates permutations *p*-value for statistical significance. Same is below.

Soil factors affecting the soil fungal community abundance were different at each developmental stage of the plantation forests; thus, Mantel test analyses were used to identify soil factors affecting soil fungal community abundance at each developmental stage. We found that soil fungal community abundance was strongly influenced by TK at the young forest stage (r = 0.99 **, Figure 9A); and TN had the greatest influence on soil fungal community abundance at the near-mature forest stage (r = 0.99 **, Figure 9B), and SOC, AN, and S-SC at the over-ripe forest stage (r = 0.49 **, Figure 9C). The soil bacterial community abundance in the mixed forest was most affected by AN (r = 0.99 **, Figure 9D).

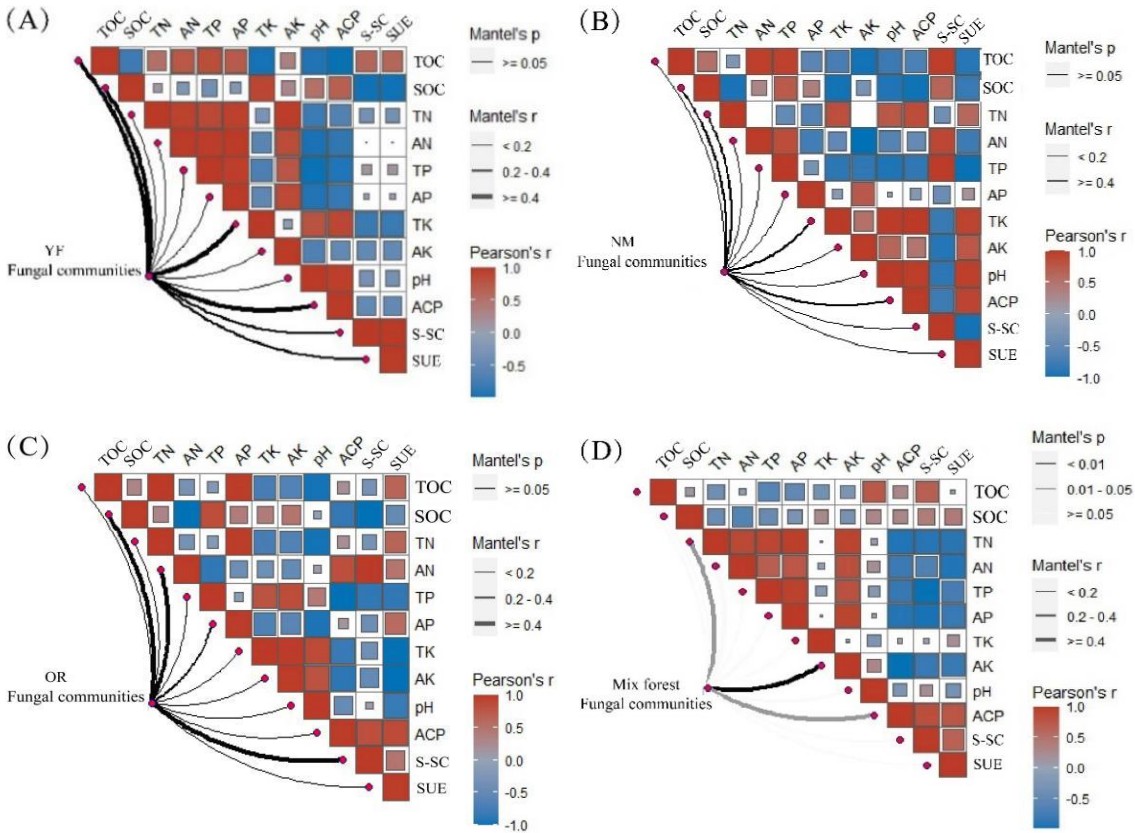

**Figure 9.** Identification of key driving factors affecting fungal community structure in different forest ages and stand types. (**A**) Correlation of soil bacterial communities with soil nutrients and enzyme activities at the young forest; (**B**) Correlation of soil bacterial communities with soil nutrients and enzyme activities in near-mature forests; (**C**) Correlation of soil bacterial communities with soil nutrients and enzyme activities in overripe forests; (**D**) Correlation of soil bacterial communities with soil nutrients and enzyme activities in Chinese fir–*P. bournei* mixed forests. Note: Significant correlation at Mantel's r ≥ 0.4. Correlation is significant at 0.2 ≤ Mantel's r ≤ 0.4. Data: means ± SD (*n* = 3). Color gradient indicates Spearman's correlation coefficient; the side width corresponds to the correlation coefficient for distance calculated by Mantel; color shade indicates permutations *p*-value for statistical significance.

## 4. Discussion

### 4.1. Analysis of the Effects of Forest Age and Stand Type on Rhizosphere Soil Chemical Properties and Enzyme Activity

We observed a continuous increase in TOC and SOC as the forests aged from 10 to 45 years old (Table 1). There is a similar trend to those reported by Deng et al. and Yu et al., who found that SOC concentration continued to increase with the increase in Chinese fir plantation age [3,46]. However, there are also many conclusions inconsistent with the results of this study. For example, Wei et al. found that SOC content decreased significantly in the near-mature forest stage and increased slightly in the over-mature forest stage [47]. Chen et al. found that SOC content continued to decline with the increasing forest age [48]. In this study, the TK and AK contents showed the same trend as SOC and varied significantly among different ages. TP, AP, TN, and AN all declined to the lowest trend in the near-mature forest stage and varied significantly. These observations are consistent with the research by Liu et al., who found that rhizosphere soil N and P contents changed in a V-shaped pattern with increasing forest age [49].

The initial low C content may be due to the fact that a large amount of wood was taken from the forest floor at the end of the last harvest period through slash burning, which caused a large amount of C to be burned off [50]. With the increasing forest age, the understory detritus continues to accumulate; however, the decomposition rate is slow and soil C gradually accumulates. Thus, the soil nutrients (N and P) consumed by trees before 25–30 years will exceed the returned soil nutrients [51,52]. This result can also be explained by the reduction in understory vegetation, which is related to the premature closure of the canopy during the growth of trees [48,53]. We observed a continuous decrease in understory vegetation from the young forest to near-mature forest, which supports the above explanation.

The K content increased with the increasing stand age. The number of plant roots and the amount of soil microorganisms increased, which promoted the conversion of insoluble mineral state potassium to water-soluble, exchangeable potassium. Moreover, the increase in AMF promoted the accumulation of K. In addition, Zhang Yuan et al. [54] found, in their study on gooseberry, that the demand for fast-acting K was higher during the juvenile period of plant growth, which may also be the cause of the younger stands having opposing trends in pH and K. In a study by Li et al. [55], it was found that certain phosphorus-solubilizing bacteria existed in the inter-root soil of cedar, and that in the long-term phosphorus-deficient environment of cedar, the content of phosphorus-solubilizing bacteria was higher. Moreover, phosphate-solubilizing bacteria can produce acidic substances and reduce soil pH.

The soil nutrient content and soil pH of the mixed forest were significantly higher than those of the planted forest. Zhongming et al. [56] and Joshi B [57] had similar results as ours; thus, it is recommended to construct a mixed forest while extending the rotation period.

### 4.2. Rhizosphere Soil Microbial Diversity and Abundance Influenced by Tree Age and Stand Type

In this study, after the introduction of *P. bournei* into a Chinese fir plantation, the α-diversity of the soil microbial community was greatly affected, which was manifested as the influence of the α-diversity of the soil bacterial community being greater than that of the soil fungal community (Tables 2 and 3). Moreover, the soil microbial community structure was obviously separated (Figures 4 and 5). Diakhate et al. found that bacterial α-diversity in millet and peanut intercropping was significantly affected by the planting pattern, which was consistent with the results of this study [58]. Similarly, Debenport et al. found that the diversity of bacterial communities increased significantly after the introduction of *P. reticulatum* or *G. senegalensis* into millet [59]. One of the reasons for this result is that bacterial communities are more sensitive than fungal communities in the early stages of stand-type change [60]. Studies have shown that in the early stages of forest development, environmental changes have a greater impact on soil bacterial communities, and soil bacterial communities change faster and are more sensitive than fungal

communities [61,62]. Another reason is that the change in microbial species richness and diversity is not necessarily affected by the change in the structure of microbial communities, because changes in certain groups may be compensated by changes in other groups [63,64].

In *C. lanceolata* plantation forests of different stand ages, the quantity of OTUs of the soil bacterial community, first increased and then decreased, as stands aged (Tables 1 and 2). This may be due to the fact that controlled burning before afforestation caused the biomass that was not taken out of the forest floor to return to the soil, and after entering the fast-growing stage, the soil nutrients declined rapidly, and *C. lanceolata* continuously released organic acids through the root system as a way to enhance bacterial activity in order to obtain sufficient nutrients [15,65]. This is also corroborated by the fact that both acid phosphatase and urease activities reached their maximum at the near-mature forest stage. This is consistent with the monotonic increase in urease activity over 15–35 years of redwood plantations, as reported by Kang et al. [66].

The abundance of the fungal community continued to decline with stand age, with diversity being lowest at the near-mature forest stage (Tables 1 and 3), which may result because of two reasons. First, because of the severe soil acidification due to acidic substances produced by lower nutrient content *C. lanceolata* root secretions and microbial decomposition of difficult-to-decompose materials, the abundance of fungal species intolerant of acidic environments declined [67,68]. Second, as it grows, the forest canopy closes, light decreases, water and heat conditions within the forest become poorer, and the diversity of understory vegetation decreases. This is consistent with the results of Zhang et al. [69].

### 4.3. Effect of Stand Age and Stand Type on the Keystone Taxa

We identified site-specific OTUs in pure and *C. lanceolata* × *P. bournei* forests (Tables 1–3), which may serve as key taxa to explain the variation in microbial communities between plantations. At the level of bacterial phylum, *Chloroflexi* and *Firmicutes* were higher in pure and mixed stands of fir (Figure 1). It has been suggested that *Chloroflexi* and *Firmicutes* may play an important role in participating in carbon and nitrogen cycling, using recalcitrant carbon fixation [70,71], and have the ability to degrade and fix nitrogen from organic matter, lignin, and cellulose [72,73]. However, the abundance of *Chloroflexi*, *Firmicutes*, and other strains did not differ significantly in pure and mixed Chinese fir–*P. bournei* forests. This may be because during the dry season, the amount of apoplankton increased in the pure Chinese fir forest, and the dissolved oxygen content in the soil increased, which favored the growth of aerobic microorganisms and higher microbial biomass.

As for fungi, the introduction of broad-leaf trees not only increased the diversity of fungal communities but also significantly increased the relative abundance of *Basidiomycota* compared to pure stands of Chinese fir. Although previous studies confirmed that *Ascomycota* is more capable of hydrolyzing cellulose than *Basidiomycota* [74], we found no difference in *Ascomycota* abundance between pure and mixed stands of Chinese fir. Thus, *Basidiomycota* played a greater role in mixed stands than in pure stands (Figures 3 and 4b). The study by Y. Liang et al. found that *Basidiomycota* often formed mycorrhizae in symbiosis with plants to promote nutrient uptake and plant growth [75]. Therefore, the higher abundance of Basidiomycota in the mixed forest may be due to the higher number of symbiotic bacteria in the mixed forest.

### 4.4. Different Stand Age and Forestry Management Practices Might Influence Soil Microbial Communities via Soil Properties

Most of the studies considered soil factors and vegetation as the main factors affecting microbial community structure. Nielsen et al. [76] found that changes in soil properties had strong effects on bacterial and archaeal communities. Soil pH is the main factor affecting soil bacterial diversity [77,78]. The bacterial community diversity showed an upward then downward trend in different stand ages of Chinese fir plantations, and according to the Mantel test analysis, soil pH was significantly correlated with bacterial community diversity and had the greatest effect on bacterial community (r = 0.99 **). These findings

may be due to the increased nutrient stress on Chinese fir trees as they aged, which releases root secretions and organic acids. These secretions have an attractive effect on some microorganisms, and these chemotactic bacteria can accumulate and multiply in large numbers in the inter-rhizosphere. For example, Li et al. found that when intercropping between grasses and legumes, the roots of legumes such as broad beans (*Vicia faba*) acidified the soil by releasing organic acids and proteins, which activated insoluble phosphorus and thus promoted phosphorus uptake by grasses such as maize (*Zea mays*) [79]. This process also resulted in no significant difference in bacterial community diversity between *C. lanceolata* plantation and *C. lanceolata–P. bournei* mixed forests at the near-mature stage. Liu et al. showed that the loss of soil bacterial diversity was mainly due to the decrease in soil pH [80]. Thus, bacterial diversity decreases in the overripe forest stage.

There are two main reasons affecting the diversity of soil fungal communities in fir plantations of different stand ages. One reason is soil nutrients. According to the Mantel test analysis, the fungal community diversity at different stand ages was initially the most influenced by potassium (r = 0.99 **) and gradually shifted to nitrogen (r = 0.49 **). Maisto et al. [81] showed that forest soil nutrients are mainly derived from apoplastic matter. The apoplastic material of Chinese fir plantations is mainly needles, which contain a large amount of lignin, cellulose, and tannin, which have relatively high C/N, leading to soil N deficiency and inhibition of fungal activity [82]. The other explanation is soil pH. Although pH did not have a direct effect on fungal community diversity according to the Mantel test analysis, this may be because the decrease in soil pH drives aluminum toxicity, which is detrimental to microbial growth [83].

Soil pH has a significant effect on soil microbial communities. In plantation forests, soil bacterial diversity tended to increase and then decrease with a gradual increase in soil acidity, a result that is consistent with the findings of F. Wu et al. [84]. Fungal diversity decreased significantly from the young stage to the near-mature stage and slightly increased after the overripe stage. The analysis was related to the gradual increase in soil acidity and the gradual decrease in nutrients in the plantation forest [85], which was consistent with the results of Dennis [86]. In mixed forests, soil acidity is maintained in a weakly acidic state, and the broad-leaf tree *P. bournei* may improve the quality of litter inputs, while microorganisms can participate in nutrient cycling through litter decomposition, leading to higher decomposition rates and more nutrients being released into the soil [87], which in turn affects forest ecology. This process is consistent with the results reported by Xia et al. on the introduction of Elaeocarpus decipiens Hemsl and Michelia macclurei broad-leaf trees into Chinese fir monoculture plantations, where the growth of Chinese fir was promoted and accompanied by changes in soil microorganisms [88].

## 5. Conclusions

From the perspective of microbiology, the results of this study are presented. The introduction of *P. bournei* into a Chinese fir plantation did not affect the number and diversity of bacterial communities in rhizosphere soil under a single planting pattern, but the diversity of fungal communities was twice that of the Chinese fir plantation. Rhizosphere soil nutrients, acid phosphatase, and sucrase were also significantly higher than in the Chinese fir plantation. Although the soil nutrients increased slightly at the over-ripening stage, there was still a considerable gap compared with the mixed forest. Therefore, in order to ensure the sustainable development of Chinese fir plantations, more attention should be paid to the middle and late stages of the growth of Chinese fir plantations rather than the early stages. In addition, the construction of mixed forest models should be used as the main forestry management method.

**Author Contributions:** Conceptualization, W.L. and J.J.; methodology, W.L., M.C. and J.J.; software, W.L. and H.S.; resources, L.W. and X.F.; writing—original draft preparation, W.L. and H.S.; writing—review and editing, W.L., H.S. and J.J.; visualization, W.L. and H.S.; supervision, J.J.; project administration, W.L. and J.J. All authors have read and agreed to the published version of the manuscript.

**Funding:** This study was supported by the National Natural Science Foundation of China (32071612) and the Baishanzu National Park Scientific Research Project (2022JBGS03, 2021ZDLY01).

**Data Availability Statement:** Data available on request due to restrictions surrounding, e.g., privacy or ethics. The data presented in this study are available on request from the corresponding author.

**Conflicts of Interest:** The authors declare no conflict of interest.

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
