# Peer review of "Diversity and Structure of Soil Microbial Communities in Chinese Fir Plantations and Cunninghamia lanceolataPhoebe bournei Mixed Forests at Different Successional Stages"

_forests, doi:10.3390/f14101977_

Round 1

Reviewer 1 Report

Dear,

After reading the article Differences in the structure and function of soil microbial communities in different aged Cunninghamia lanceolata plantations and Cunninghamia lanceolata - Phoebe bournei mixed forests I have a few comments.

The topic is currently very popular, and there are many papers considering the soil microbial community based on the 16S and ITS rRNA gene sequencing. In this paper the efforts have been made to connect the community structure and soil properties.

The paper contains valuable data and deserves to be published.

However, some issues should be resolved before publication.

Please check, whether the names of prokaryotes were given according to the valid publication of the phyla.

Methodology for soil analysis should be given in more detailed way, at least as reference (subheading 2.2. Soil Properties and Soil Enzyme Activities Analysis).

Best regards,

Author Response

[Comment 1] Please check, whether the names of prokaryotes were given according to the valid publication of the phyla.

ANSWER:

Done as suggested. All microbial phylum level classification names and related species functions are from published literature and reference books. For example, L294 – 297, Although previous studies confirmed that Ascomycota is more capable of hydrolyzing cellulose than Basidiomycota [74], we found no difference in Ascomycota abundance between pure and mixed stands of Chinese fir. Thus, Basidiomycota played a greater role in mixed stands than in pure stands (Fig 3 and 4b).

[74] Yelle, D.J.; Ralph, J.; Lu, F.; Hammel, K.E. Evidence for cleavage of lignin by a brown rot basidiomycete. Environ. Microbiol. 2008, 7, 1844-1849.

[Comment 2] Methodology for soil analysis should be given in more detailed way, at least as reference (subheading 2.2. Soil Properties and Soil Enzyme Activities Analysis).

ANSWER:

Done as suggested. According to the problem, the description of all test indicators has been rewritten, including soil physical and chemical properties, enzyme activity, and microbial high-throughput sequencing. For details, see L140 - L235 please.

Reviewer 2 Report

In the article, the authors studied the effect of stand age and stand type on soil fertility and microbial diversity, composition, and structure in Chinese farm plantations. This topic is relevant because there are few reports on the impact of changes in plantation type on soil characteristics, as well as on the further impact on the diversity, composition and abundance of microbial (bacteria and fungi) communities and the identification of key driving factors (soil chemical factors and enzyme activities) influencing on soil microbial communities during single and mixed cultivation of Chinese fir (C.lanceolata × P.bournei).

The work is relevant, but there are a number of comments:

1. The signatures in Figures 2, 4 are not very readable.

Best regards, reviewer.

Author Response

ANSWER to Reviewer #2:

 [Comment 1] The signatures in Figures 2, 4 are not very readable.

ANSWER:

Done as suggested. Figures 2 and 4 have been reproduced to enhance clarity. See L74 and L106, please.

Reviewer 3 Report

In general, the authors have done a good job explaining the background information necessary to appreciate the rationale and results of the experiments. The manuscript was prepared correctly. Methodology and analysis of results rather don't raise any objections.
However, some minor amendments are needed. The discussion of the results is written a bit generally. There are papers related to this topic that the authors did not cite. An assessment putting the findings into perspective and make a solid conclusion is missing. The authors should emphasize more the novelty and usefulness of the results.

Author Response

ANSWER to Reviewer #3:

[Comment 1] There are papers related to this topic that the authors did not cite.

ANSWER:

In this paper, relevant references are cited, such as the 6th-8th in the L59 ([8] Zhao, J.Z.; Xie, D.M.; Wang, D.Y.; Deng, H.B. Current status and problems in certification of sustainable forest management in China. Environ Manage. 2011, 6, 1086-1094. [9] Wang, W.F.; Wei, X.H.; Liao, W.M.; Blanco, J.A.; Liu, Y.Q.; Liu, S.R.; Liu, G.H.; Zhang, L.; Guo, X.M.; Guo, S.M. Evaluation of the effects of forest management strategies on carbon sequestration in evergreen broad-leaved (Phoebe bournei) plantation forests using FORECAST ecosystem model. For. Ecol. Manag. 2013, 300, 21-32.), the 10th in the L66 ([10]Khlifa, R.; Paquette, A.; Messier, C.; Reich, P.B.; Munson, A.D. Do temperate tree species diversity and identity influence soil microbial community function and composition? Ecol. Evol. 2017, 19, 7965-7974.), and the 18th-19th in the L73([18] Zhang, P.; Guan, P.; Hao, C.; Yang, J.; Xie, Z.; Wu, D. Changes in assembly processes of soil microbial communities in forest-to-cropland conversion in Changbai Mountains, northeastern China. Sci. Total Environ. 2022, 818, 151738. [9] Meng, M.J.; Lin, J.; Guo, X.P.; Liu, X.; Wu, J.S.; Zhao, Y.P.; Zhang, J.C. Impacts of forest conversion on soil bacterial community composition and diversity in subtropical forests. Catena. 2019, 175, 167-173.). The effects of single plantation on soil properties and soil microbial communities after conversion into mixed forests are described.

[Comment 2] An assessment putting the findings into perspective and make a solid conclusion is missing.

ANSWER:

Done as suggested. According to the requirements of experts, the conclusion has been rewritten. Specifically, as follows. From the perspective of microbiology, combined with the results of this study. The introduction of P.bournei into Chinese fir plantation did not affect the number and diversity of bacterial communities in rhizosphere soil under single planting pattern, but the diversity of fungal communities was twice that of Chinese fir plantation. Rhizo-sphere soil nutrients and acid phosphatase, sucrase were also significantly higher than Chinese fir plantation. Although the soil nutrients increased slightly at the over-ripening stage, there was still a considerable gap compared with the mixed forest. Therefore, in order to ensure the sustainable development of Chinese fir plantations, more attention should be paid to the middle and late stages of the growth of Chinese fir plantations rather than the early stages. In addition, the construction of mixed forest models should be used as the main forestry management method.

Reviewer 4 Report

Dear authors,

these are my suggestions to make the manuscript better.

line 26- Please eliminate “The”.

line 27- Please, separate the words.

line 37- Please, separate the words.

line 59- Please eliminate “.”

line 75- The sentence seems incomplete.

line 81- Please eliminate “However”.

lines 90-99- This part is not not clear. Please check the grammar.

line 206- bacterial/fungal

line 224- Table 1.- Please eliminate “.”

line 77-  According to which taxonomy and systematics was this conclusion written? There are many different systematics of fungi. Which systematics was used here?

line 212- Please, separate the words.

line 217- italic

line 228- This part is written twice.

lines 236-240- This part is not not clear. Please check the grammar.

line 245- Table 1, 3 Please, separate the words.

line 255- Firmicutes were, Please, separate the words.

lines 280-283- This part is not not clear. Please check the grammar.

line 311- This part is not not clear. Please check the grammar.

Moderate editing of the English language is required.

Author Response

Response to reviewers

ANSWER to Reviewer #4:

[Comment 1] line 26- Please eliminate “The”.

ANSWER:

Done as suggested.

[Comment 2] line 27- Please, separate the words.

ANSWER:

Done as suggested.

[Comment 3] line 37- Please, separate the words.

ANSWER:

Done as suggested.

[Comment 4] line 59- Please eliminate “.”

ANSWER:

Done as suggested.

[Comment 5] line 75- The sentence seems incomplete.

ANSWER:

Here, we want to talk about what we should know about microbial species, which is very helpful in understanding the effects of changes in soil properties and forest types on microbial communities.

[Comment 6] line 81- Please eliminate “However”.

ANSWER:

Done as suggested.

[Comment 7] lines 90-99- This part is not clear. Please check the grammar.

ANSWER:

Done as suggested. Check and revise this paragraph according to the question.

[Comment 8] line 206- bacterial/fungal

ANSWER:

Done as suggested. Delete and re-describe the paragraph according to the question.

[Comment 9] line 224- Table 1.- Please eliminate “.”

ANSWER:

Done as suggested.

[Comment 10] line 77-  According to which taxonomy and systematics was this conclusion written? There are many different systematics of fungi. Which systematics was used here?

ANSWER:

The method used in this paper is ITS1 to classify fungi. The primers are 5 ' -GCTGCGTTCTTCATCGATGC-3 ', 5 ' -CTTGGTCATTTAGAGGAAGTAA-3 '. The specific test process and the primers used are shown in 2.3.

[Comment 11] line 212- Please, separate the words.

ANSWER:

Done as suggested.

[Comment 12] line 217- italic

ANSWER:

Done as suggested.

[Comment 13] line 228- This part is written twice.

ANSWER:

Done as suggested. Duplicates have been removed and re-described

[Comment 14] lines 236-240- This part is not clear. Please check the grammar.

ANSWER:

Done as suggested.

[Comment 15] line 245- Table 1, 3 Please, separate the words.

ANSWER:

Done as suggested.

[Comment 16] line 255- Firmicutes were, Please, separate the words.

ANSWER:

Done as suggested.

[Comment 17] lines 280-283- This part is not clear. Please check the grammar.

ANSWER:

Done as suggested.

[Comment 18] line 311- This part is not clear. Please check the grammar.

ANSWER:

Done as suggested.

[Comment 19] Moderate editing of the English language is required.

ANSWER:

The text has been revised by native English speakers and carefully corrected.

Round 2

Reviewer 3 Report

The MS has been corrected according to the Reviewer’s comments and is suitable for publication in its present form.